

# Antimicrobial activity of *Androgaphis paniculata* intimate wash and commercial formulations against vaginal pathogens using the broth microdilution method

Nur Rina Alissa Razian[1], Tuan Noorkorina Tuan Kub[1,2], Nurdini Afiqah Suhaimi[3], Tuan Nadrah Naim Tuan Ismail[4], Muhamad Alojid Anis Amiera[1], Gayus Sale Dafur[1,5], Fatmawati Lambuk[6], Rohimah Mohamud[6], Ramlah Kadir[6], Norzila Ismail[7] and Norhayati Yusop[4]

[1] Department of Medical Microbiology & Parasitology, School of Medical Sciences, Universiti Sains Malaysia, Kota Bharu, Kelantan, Malaysia

[2] Medical Microbiology & Parasitology Laboratory, Hospital Pakar Universiti Sains Malaysia, Kota Bharu, Kelantan, Malaysia

[3] Faculty of Applied Sciences, Universiti Teknologi MARA, Shah Alam Campus, Shah Alam, Selangor, Malaysia

[4] School of Dental Sciences, Universiti Sains Malaysia, Kota Bharu, Kelantan, Malaysia

[5] Department of Biology, Federal University of Education, Pankshin, Plateau State, Nigeria

[6] Department of Immunology, School of Medical Sciences, Universiti Sains Malaysia, Kota Bharu, Kelantan, Malaysia

[7] Department of Pharmacology, School of Medical Sciences, Universiti Sains Malaysia, Kota Bharu, Kelantan, Malaysia

Corresponding author
Tuan Noorkorina Tuan Kub,
tnkorina@gmail.com

## ABSTRACT

**Background.** This study aims to evaluate the antimicrobial efficacy of a newly formulated *Androgaphis paniculata* intimate wash in comparison with commercial brands (SebaMed®, Sumber Ayu, Lactacyd®, and Good Virtue & Co.) against vaginal pathogens. The antimicrobial activity of each formulation was tested against *Streptococcus agalactiae*, *Escherichia coli*, and *Candida albicans*. Additionally, *Lactobacillus crispatus* was tested to ensure the formulations did not inhibit normal vaginal flora.

**Methods.** The broth microdilution method was used to assess antimicrobial efficacy by calculating the percentage of microbial inhibition. Minimum inhibitory concentration (MIC) was determined for *A. paniculata* and commercial intimate washes against the tested pathogens, except for *L. crispatus*, due to the inability to observe turbidity. Minimum bactericidal concentration (MBC) was evaluated for the *A. paniculata* intimate wash against the same pathogens.

**Results.** The *A. paniculata* intimate wash showed the highest percentage of microbial inhibition across all tested pathogens. It also recorded MIC and MBC values of 3.125 mg/ml against *E. coli* and *C. albicans*. Among commercial products, SebaMed® demonstrated the strongest antimicrobial effect against *S. agalactiae* at the same MIC concentration. The study concluded that the *A. paniculata* intimate wash is a potentially effective treatment for selected vaginal pathogens at higher concentrations and may promote the growth of *L. crispatus*. SebaMed®, a clinically established product, also exhibited notable antimicrobial properties. However, further *in vivo* and clinical trials are necessary to confirm efficacy and assess toxicity. The study highlights the need for

continued research into the potential of herbal intimate washes for treating vaginal infections.

# INTRODUCTION

A healthy vagina is dominated by lactobacilli species and a small presence of fungal species. The vaginal microbiota can change throughout life and can be influenced by menstruation (*Gajer et al., 2012*), vaginal douching (*Sabo et al., 2019*), antibiotic use (*Mayer et al., 2015*), and hormonal fluctuations (*Huang et al., 2014*). *L. crispatus* supports vaginal stability by providing probiotic benefits, including reduced risk of human immunodeficiency virus (HIV) and sexually transmitted diseases (STDs), enhanced antibiotic sensitivity, and inhibition of pathogens (*Li et al., 2021*).

However, vaginal dysbiosis can be triggered by stress, antibiotics, or sexual activity, resulting in decreased *Lactobacillus* spp. and elevated vaginal pH. Some infections can be exposed to opportunistic overgrowth of *E. coli* and *Gardnerella vaginalis* which lead to bacterial vaginosis (BV) (*Holdcroft, Ireland & Payne, 2023*). BV can be detected through common symptoms such as fishy odors, grayish-white discharge, and increased vaginal pH (>4.5) (*Abdul-Aziz et al., 2019*). Vulvovaginal candidiasis (VVC) is caused by *C. albicans*, leading to itching, dyspareunia, oedema, discomfort, and dysuria (*Nyirjesy et al., 2022*). In addition, urinary tract infections (UTIs) are caused by uropathogenic *E. coli* that can enter the urinary tract and it can be detected through some symptoms, including fever, burning, itching, blistering in the vaginal area, suprapubic discomfort, and pyuria (*Kaur & Kaur, 2021*).

Metronidazole is an approved antibiotic to treat BV. It works well to prevent the infection since anaerobic bacteria can metabolize it into nitroso radicals, which can damage microbial DNA and induce cell lysis (*Verwijs et al., 2020*). However, this treatment often caused side effects and led to antibiotic resistance (*Fan et al., 2024*) since most of the 50 strains of *G. vaginalis* were resistant to metronidazole (*Abbe & Mitchell, 2023*).

Fluconazole and ciprofloxacin are commonly used for treating VVC and UTIs. Fluconazole, the first-line antifungal for *C. albicans*, is now facing reduced efficacy due to emerging azole-resistant strains (*Costa-De-Oliveira & Rodrigues, 2020*). Ciprofloxacin, a broad-spectrum fluoroquinolone, has also seen declining effectiveness due to increasing resistance (*Reis et al., 2016*). These challenges underscore the need for traditional alternative therapies.

Vaginal rejuvenation was used to restore the vaginal tissues. Lubricants, hormone replacement therapy, and Kegel exercises, are also commonly used to treat vaginal issues (*Karcher & Sadick, 2016*). However, as these approaches are expensive and cause side effects, herbal plants offer a promising alternative due to their potential therapeutic benefits. For example, cyclamen is used to treat vaginal infections and infertility, and it can also treat vaginal discharge (*Turan et al., 2019*).

In this study, it is important to evaluate antiseptic concentrations as antibacterial activity against *Lactobacillus* spp. can disrupt the vaginal ecosystem. Therefore, a study found that 2% w/v ethanolic red *Piper betle* extract is effective without significantly affecting *Lactobacillus* spp., which support it as vaginal antiseptics (*Kusuma, Tjitraresmi & Susanti, 2017*). These findings support the evidence that herbal medications have significant potential in treating vaginal infections.

Nowadays, formulation of intimate wash should protect against vulvovaginal infections and dryness, and be clinically tested for the vulvovaginal area (*Murina et al., 2023*). However, their excessive usage has been increasingly associated with antimicrobial resistance (*Caioni et al., 2023*). Research has shown that triclosan, which is widely used in industrial products, can lead to increased estrogen activity and disrupt the body's hormonal balance (*Maksymowicz et al., 2022*). Similarly, sodium lauryl sulfate, which is used as surfactants, can disrupt skin barrier integrity, alter the composition of the microbiota, and increase the risk of infection. The need for antibiotic treatment can contribute to the growing issue of antimicrobial resistance (*Leoty-Okombi et al., 2021*). Therefore, both ingredients highlight the importance of evaluating the safety in intimate wash formulations.

There is an interest in herbal plant-based alternatives. *A. paniculata*, which contains andrographolide may act synergistically and exhibit a lower tendency to induce resistance. In the context of vaginal health, *A. paniculata* has been used for treating infections and inflammation, making it suitable for intimate wash formulations (*Tyagi et al., 2025*). Despite the growing use of herbal vulvar cleansers, studies on the antimicrobial efficacy of *A. paniculata* against vaginal pathogens remain limited. To date, only *Srivastava et al. (2024)* has reported its activity in a soap formulation against skin pathogens such as *E. coli*, *Staphylococcus aureus*, *Pseudomonas aeruginosa*, and *Bacillus licheniformis*.

The novelty of the study lies in the formulation, accompanied by evaluation of its antimicrobial activity. This study supports the development of safer, plant-derived alternatives to conventional synthetic intimate wash. *A. paniculata* is highly explored as it has been employed to treat cancer, diabetes, bronchitis, and malaria. Its bioactive compounds are known for their antimicrobial, anti-inflammatory, and anti-diabetic properties (*Okhuarobo et al., 2014*). A study by *Radhika & Lakshmi (2010)* demonstrated that chloroform extracts from the stem and root possess antibacterial and antifungal properties comparable to benzyl penicillin. These studies highlight the potential of *A. paniculata* as a natural alternative against resistant pathogens.

*A. paniculata* exhibits a broader antimicrobial spectrum than other herbal extracts in treating vaginal pathogens. *Mathur et al. (2011)* reported that *A. paniculata* exhibited superior antifungal activity against *Aspergillus niger* and *C. albicans* compared to *Berberis aristata*, *Tinospora cordifolia*, and *Plantago depressa*. *Dedhia et al. (2018)* also demonstrated that *A. paniculata* has shown stronger antimicrobial activity against *C. albicans* and *S. aureus* compared to *Azadirachta indica* and *Curcuma longa*. In addition, according to *Anis Amiera et al. (2024)*, andrographolide has also demonstrated strong anticancer activity in gastric cancer cell lines. These findings highlight the therapeutic promise of *A. paniculata*, not only for its antimicrobial activity but also for its wider biomedical applications.

Andrographolide exerts antimicrobial effects including inhibition of microbial growth, disruption of biofilm formation, and modulation of resistance pathways. *Žiemyte et al. (2023)* mentioned that andrographolide demonstrated the inhibition against planktonic *Candida* cell growth and prevented biofilm formation. It is effective in disrupting mature biofilms and can lead to biofilm detachment. These findings were further supported by scanning electron microscopy analysis. Andrographolide was also found to inhibit biofilm formation and enhance the susceptibility of *P. aeruginosa* by downregulating the expression of the MexAB-OprM (tripartite multidrug resistance system in *P. aeruginosa*) efflux pump gene (*Zhang et al., 2021*).

Studies directly comparing synthetic intimate washes with *A. paniculata*-based herbal formulations remain limited. Some evidence somehow supports its potential as a natural alternative product. *Vangchanachai et al. (2023)* reported that patients who used a liquid soap with *A. paniculata* and perilla oil experienced reduced skin pain and better moisturization than chlorhexidine. In addition, *Plianrungsi & Kulthanaamondhita (2021)* demonstrated that *A. paniculata* mouthwash effectively reduces plaque accumulation, with fewer side effects. These findings support the potential use of *A. paniculata* as a safer natural alternative due to its antiplaque properties. Therefore, this research supports the development of safer alternatives, while also contributing to efforts in minimizing antimicrobial resistance.

The findings underscore the antimicrobial potential of *A. paniculata* for intimate wash formulation since there is a need to identify alternative or complementary strategies to antibiotics (*Mdarhri et al., 2022*). Clinical trials have further validated the efficacy of *A. paniculata* in treating respiratory tract infections. Consequently, *A. paniculata*, especially andrographolide, is considered as a candidate for drug development (*Hossain et al., 2021*). This study contributes to evidence supporting the use of medicinal plants as personal hygiene applications.

In this study, an intimate wash derived from the *A. paniculata* crude extract and four commercial intimate washes, namely SebaMed®intimate wash (Sebapharma GmbH & Co. KG, Salzig, Germany), Sumber Ayu intimate wash (Wipro Unza Sdn. Bhd., Subang Jaya, Malaysia), Lactacyd®intimate wash (Sanofi Aventis Sdn. Bhd., Petaling Jaya, Malaysia), and Good Virtue & Co. intimate wash (Citychemo Manufacturing Sdn. Bhd., Selangor, Malaysia), were tested against *C. albicans*, *E. coli*, and *S. agalactiae*. *L. crispatus* was included to assess its resistance to *A. paniculata* and commercial intimate washes. A study by *Mändar et al. (2023)* selected specific strains based on their proven probiotic potential and safety, while *Er et al. (2019)* confirmed its activity through lactic acid, hydrogen peroxide, and proteolytic enzyme production. Antimicrobial activity was evaluated to determine the percentages of microbial inhibition, minimum inhibitory concentration (MIC) and minimum bactericidal concentration (MBC) of the pathogens. This study hypothesizes that the *A. paniculata* intimate wash will show stronger antimicrobial effects against pathogens while preserving *L. crispatus* growth, compared to commercial products.

## MATERIALS AND METHODS

### Preparation of *A. paniculata* extract

The maceration method was used for crude extraction of *A. paniculata* (*Patel, Panchal & Ingle, 2019*). One litre of absolute ethanol (solvent) was used to soak 100 g of *A. paniculata* powder and heated at 30 °C overnight on a C-MAG HS 7 hot plate stirrer (IKA, India). The following day, the mixture was filtered out using Whatman™, CAT No. 1001-150 sterile filter paper (Merck KGaA, Darmstadt, Germany) and the compounds that remained on the filter paper were disposed of. The filtration was then evaporated using a BUCHI rotary vaporizer (BÜCHI Labortechnik AG, Swiss, Switzerland) until it became a paste form and stored at 2 to 5 °C.

### Formulation of *A. paniculata* intimate wash

The paste form extract was then diluted to prepare 3.125 mg/ml of *A. paniculata* intimate wash. The extract was initially diluted with 1% dimethyl sulfoxide (DMSO) and mixed with a fragrance agent (peppermint oil), solubilizing agent (Tween 20), preservatives (potassium sorbate and sodium benzoate), surfactants (Sodium Coco-Glucoside Tartrate, Cocamidopropyl Betaine and Sodium Cocoyl Hydrolyzed Wheat Protein) and sterile deionized water as solvent (adapted from *Naimat et al., 2021*). Citric acid was also added to achieve a final pH 4.0. All procedures were performed aseptically to ensure sterility.

### Preparation of diluted commercial intimate washes

Four commercial intimate washes, SebaMed®, Sumber Ayu, Good Virtues & Co., and Lactacyd®, were diluted to 3.125 µl/ml by mixing 25 µl of each intimate wash with eight ml sterile distilled water to match the concentration of the *A. paniculata* formulation (3.125 mg/ml). Assuming 100% initial concentration, this standardized dilution enabled consistent *in vitro* comparison of antimicrobial activity. Although not reflective of real-world usage, it ensured controlled testing conditions (*Diaz et al., 2011*; *Barata et al., 2023*).

### Bacterial and fungal stock culture preparation

*Candida albicans* (ATCC®90028™), and *Escherichia coli* (ATCC®25922™) were sourced from the American Type Culture Collection (ATCC, USA) as standardized reference strains and were used without re-identification. In contrast, *Streptococcus agalactiae* (strain ID: 23041118) and *Lactobacillus crispatus* (strain ID: 22027717) were obtained from the Microbiology Laboratory Stock Culture at Universiti Sains Malaysia underwent confirmation using an automated identification machine, VITEK®2 (bioMérieux, Inc., Marcy-l'Étoile, France), which demonstrated an accuracy exceeding 98%. All the strains are preserved under appropriate storage conditions in the departmental microbial culture repository for future references and research use.

*S. agalactiae* and *E. coli* were subcultured on OXOID Columbia Sheep Blood Agar (Thermo Fisher Scientific, Massachusetts, USA), while *C. albicans* was subcultured on OXOID Sabouraud Dextrose Agar (Thermo Fisher Scientific, Massachusetts, USA) and they were incubated aerobically for 18 to 24 h at 37 °C. *L. crispatus* was subcultured on OXOID Columbia Sheep Blood Agar (Thermo Fisher Scientific, Massachusetts, USA) and incubated anaerobically for 48 h at 37 °C.

For antimicrobial study, OXOID Mueller Hinton Broth (MHB) (Thermo Fisher Scientific, Massachusetts, USA) was used for *E. coli*, *S. agalactiae* and *L. crispatus*, while Gibco™ RPMI Medium 1640 (Thermo Fisher Scientific, Massachusetts, USA) was used for *C. albicans*. They were incubated at 37 °C for 18 h (*S. agalactiae* and *E. coli*), 24 h (*C. albicans*) and 48 h (*L. crispatus*). All microbial suspensions were set to the 0.5 McFarland standard.

### Broth microdilution method for *A. paniculata* intimate wash

Freshly subcultured isolate was suspended in three ml of broth to reach the 0.5 McFarland standard. 200 µl of *A. paniculata* intimate wash with an initial concentration of 3.125 mg/ml was introduced in the first well. The following wells that had already been prepared with 100 µl of MHB received 100 µl of *A. paniculata* intimate wash that had been diluted from the first well (*David et al., 2021*). For *C. albicans*, RPMI 1640 was used instead of MHB. They were serially diluted to obtain concentrations of 1.563, 0.781, 0.391, 0.195, 0.098, 0.049, 0.024, 0.012, 0.006, 0.003 and 0.0015 mg/ml. Following that, 100 µl of the pathogens or *L. crispatus* suspensions were added to each well.

100 µl of MHB or RPMI was mixed with 100 µl of *A. paniculata* intimate wash for the negative control (NC). 100 µl of MHB or RPMI was added to 100 µl of microbial suspension for viability control (VC), and 200 µl of MHB or RPMI was added for sterility control (SC). The experiments were conducted in triplicates for all microbes. After the incubation, the microplates were read on a VersaMax™ microplate reader (Molecular Devices, San Jose, USA) at a wavelength of 620 nm. The percentage of inhibition was calculated based on the optical density (OD) value measured.

### Broth microdilution method for commercial intimate washes

200 µl of 3.125 µl/ml freshly diluted commercial intimate wash was added into the first well. The next wells were filled up with 100 µl of MHB or RPMI for *C. albicans*. Commercial intimate wash was serially diluted to obtain concentrations of 3.125, 1.563, 0.781, 0.391, 0.195, 0.098, 0.049, 0.024, 0.012, 0.006, 0.003 and 0.0015 µl/ml. After that, 100 µl of microbial suspension with 0.5 McFarland standards were added to each well to obtain a final volume of 200 µl (*David et al., 2021*).

Same as the protocol for *A. paniculata* intimate wash, 100 µl of MHB or RPMI was mixed with 100 µl of intimate wash for the NC. 100 µl of MHB or RPMI was added to 100 µl of microbial suspension for VC, and 200 µl of MHB or RPMI was added for SC. The experiments were repeated in triplicates. After the incubation, the microplates were read on a VersaMax™ microplate reader (Molecular Devices, San Jose, USA) at a wavelength of 620 nm. The percentage of inhibition was calculated based on the OD value measured.

### Determination of percentage of microbial inhibition

Antimicrobial efficacy of the formulated *A. paniculata* and commercial intimate washes were assessed using the broth microdilution method, with microbial inhibition percentages calculated accordingly (*Campbell, 2011*) as mentioned below. Tested control (TC), NC, VC, and SC were used to validate the accuracy, effectiveness, and aseptic integrity of the microbiological assays. NC confirms the medium does not support nontarget organisms or

show typical reactions of target microbes, while VC confirms the medium supports target growth and expected reactions. SC ensures culture media and equipment are free from contamination (*Virginia Administrative & Services Code, 2009*).

$$1 - \frac{(\text{OD of TC} - \text{OD of NC})}{(\text{OD of VC} - \text{OD of SC})} \times 100\%.$$

### Determination of MIC

The standard broth microdilution method was used to assess the antimicrobial efficacy of *A. paniculata* and commercial intimate washes by observing visible microbial growth. The lowest concentration without visible turbidity was recorded as the MIC (*Parvekar et al., 2020*). MIC values for *C. albicans*, *E. coli*, and *S. agalactiae* were determined after 24 h, while *L. crispatus* was assessed at 48 h due to its slower growth.

### Determination of MBC

To identify the optimal concentration of the *A. paniculata* intimate wash formulation, MBC determination was conducted. The microbial suspension inside first three wells were subcultured on OXOID Columbia Sheep Blood Agar (Thermo Fisher Scientific, Massachusetts, USA) (for *E. coli*, *S. agalactiae*, and *L. crispatus*), while OXOID Sabouraud Dextrose Agar (Thermo Fisher Scientific, Massachusetts, USA) was used for *C. albicans*. After the incubation, they were observed for any growth of the microbes (*Maharjan et al., 2018*).

### Data analysis

All measurements were conducted by two trained investigators. To minimize potential bias, results were independently verified by at least two researchers and any discrepancies were resolved through discussion. Statistical analysis was conducted using IBM SPSS version 29.0.1.0. The General Linear Model procedure for repeated measures was used to analyze the antimicrobial activity of the *A. paniculata* and commercial intimate washes against all the tested pathogens and *L. crispatus*. Descriptive statistics (mean $\pm$ standard deviation) were calculated for microbial inhibition percentages, and pairwise comparisons were performed to assess mean differences among intimate washes ($p < 0.05$) (*Rana, Singhal & Singh, 2013*).

## RESULTS

### Microbial inhibition percentage of vaginal pathogens

The results showed that *A. paniculata* was the most effective against these pathogens (Table 1). For *C. albicans*, *A. paniculata* intimate wash demonstrated the highest inhibition (187.08 $\pm$ 14.19%) at 0.781 mg/ml. For *E. coli*, it showed a strong inhibition (104.48 $\pm$1.77%) at 0.049 mg/ml. For *S. agalactiae*, it displayed the highest inhibition (133.93 $\pm$ 0.64%) at 0.391 mg/ml.

For the commercial intimate washes (SebaMed®, Lactacyd®, Good Virtues & Co., and Sumber Ayu), the results showed varying patterns of microbial inhibition (Table 1). For

Razian et al. (2025), *PeerJ*, DOI 10.7717/peerj.20144

**Table 1  Percentage of microbial inhibition of intimate washes against vaginal pathogens and the microbiota, *L. crispatus* using the broth microdilution method.**

| Pathogens/ microbiota | Variable (Wash) | Concentration | | | | | | | | | | | |
|---|---|---|---|---|---|---|---|---|---|---|---|---|---|
| | | 3.125 | 1.563 | 0.781 | 0.391 | 0.195 | 0.098 | 0.049 | 0.024 | 0.012 | 0.006 | 0.003 | 0.0015 |
| | | Percentage of Microbial Inhibition (Mean ± Standard Deviation) | | | | | | | | | | | |
| *C. albicans* | *A. paniculata* intimate wash | 102.06 ± 30.42 | 134.77 ± 2.80 | 187.08 ± 14.19 | 104.04 ± 20.37 | 158.97 ± 3.43 | 138.66 ± 10.44 | 115.09 ± 7.94 | 128.99 ± 5.05 | 152.86 ± 4.29 | 119.43 ± 18.50 | 111.83 ± 22.45 | 119.72 ± 12.09 |
| | SebaMed | 51.42 ± 6.79 | 56.87 ± 13.87 | 64.53 ± 10.78 | 41.99 ± 8.51 | −5.40 ± 24.28 | 7.95 ± 10.49 | 5.46 ± 7.61 | 8.48 ± 5.54 | 3.26 ± 7.21 | −1.36 ± 20.46 | 8.48 ± 5.34 | −1.31 ± 5.57 |
| | Sumber Ayu | 41.95 ± 0.20 | 19.05 ± 1.44 | 15.53 ± 4.36 | 10.66 ± 3.75 | 7.92 ± 4.12 | 0.07 ± 16.93 | −1.66 ± 6.70 | 7.62 ± 12.75 | 4.53 ± 13.03 | 16.43 ± 7.83 | 14.17 ± 5.87 | 22.56 ± 5.90 |
| | Lactacyd | 51.42 ± 5.32 | 29.90 ± 2.84 | 14.05 ± 9.89 | 5.73 ± 9.98 | 3.67 ± 10.09 | −9.03 ± 7.84 | −2.55 ± 7.85 | −0.02 ± 4.94 | 6.72 ± 4.74 | 5.90 ± 9.77 | 9.48 ± 2.47 | 8.01 ± 2.29 |
| | Good Virtues & Co. | 81.32 ± 3.67 | 72.42 ± 19.72 | 49.16 ± 22.16 | 33.05 ± 7.22 | 24.67 ± 14.95 | 19.43 ± 11.03 | 9.32 ± 23.72 | 17.04 ± 13.97 | 10.91 ± 17.86 | 12.22 ± 0.97 | 13.43 ± 0.43 | 9.03 ± 15.62 |
| *E. coli* | *A. paniculata* intimate wash | 91.56 ± 2.45 | 54.14 ± 24.47 | 82.27 ± 32.60 | 77.12 ± 37.79 | 85.15 ± 38.05 | 90.83 ± 19.15 | 104.48 ± 1.77 | 101.86 ± 0.55 | 100.61 ± 6.54 | 72.08 ± 25.99 | 73.74 ± 34.55 | 36.48 ± 11.52 |
| | SebaMed | 52.40 ± 1.79 | 35.49 ± 5.82 | 22.80 ± 2.05 | 17.45 ± 5.60 | 7.51 ± 8.29 | 4.81 ± 8.65 | 4.27 ± 8.19 | 4.16 ± 6.88 | 4.56 ± 6.40 | 4.84 ± 4.80 | 2.90 ± 4.61 | 1.96 ± 0.60 |
| | Sumber Ayu | 43.59 ± 1.05 | 21.82 ± 1.84 | 6.39 ± 4.15 | 5.11 ± 5.04 | 1.74 ± 4.55 | 2.63 ± 4.46 | −1.62 ± 0.60 | −3.26 ± 2.17 | 2.97 ± 3.87 | 0.00 ± 0.00 | 0.48 ± 4.55 | −4.44 ± 5.52 |
| | Lactacyd | 44.28 ± 1.10 | 21.63 ± 5.03 | 10.70 ± 11.42 | 5.70 ± 4.48 | 1.98 ± 5.32 | 2.42 ± 4.33 | 5.54 ± 13.45 | 0.32 ± 1.70 | −4.06 ± 5.15 | −2.85 ± 3.12 | −3.53 ± 5.15 | −3.93 ± 2.22 |
| | Good Virtues & Co. | 41.85 ± 1.16 | 27.41 ± 1.38 | 10.73 ± 9.31 | 6.42 ± 9.45 | 3.45 ± 7.25 | 3.31 ± 9.13 | 0.58 ± 7.73 | 0.51 ± 4.67 | −1.14 ± 5.08 | −1.21 ± 4.88 | 0.09 ± 8.11 | 3.38 ± 8.84 |
| *S. agalactiae* | *A. paniculata* intimate wash | 123.43 ± 7.00 | 130.23 ± 0.78 | 130.07 ± 0.50 | 133.93 ± 0.64 | 127.93 ± 0.29 | 131.07 ± 0.67 | 126.10 ± 0.62 | 128.87 ± 0.25 | 127.33 ± 0.75 | 127.03 ± 0.47 | 131.90 ± 0.95 | 130.97 ± 0.31 |
| | SebaMed | 97.31 ± 1.20 | 94.79 ± 0.86 | 29.55 ± 4.96 | 22.79 ± 7.37 | 7.84 ± 1.39 | 6.36 ± 3.62 | 6.36 ± 4.22 | 3.32 ± 2.44 | 5.38 ± 2.41 | 1.72 ± 2.60 | 6.47 ± 3.30 | −0.80 ± 3.44 |
| | Sumber Ayu | 90.07 ± 1.48 | 39.13 ± 3.59 | 28.30 ± 4.73 | 14.20 ± 2.92 | 12.08 ± 6.03 | 7.50 ± 2.82 | 4.06 ± 1.88 | 5.59 ± 1.05 | 5.17 ± 1.87 | 5.21 ± 2.86 | 7.08 ± 3.07 | 1.08 ± 5.30 |
| | Lactacyd | 96.12 ± 0.24 | 88.83 ± 5.24 | 34.57 ± 4.45 | 21.07 ± 4.40 | 15.26 ± 2.06 | 8.21 ± 3.90 | 4.61 ± 1.49 | 0.45 ± 2.27 | 4.40 ± 3.01 | 6.94 ± 1.72 | 5.05 ± 4.46 | 9.11 ± 4.23 |
| | Good Virtues & Co. | 87.88 ± 11.34 | 28.44 ± 10.70 | 31.30 ± 2.91 | 19.46 ± 5.15 | 10.88 ± 3.22 | 7.99 ± 1.38 | 13.02 ± 5.19 | 5.78 ± 2.33 | 3.75 ± 3.01 | 5.41 ± 1.38 | 8.57 ± 8.69 | 5.41 ± 4.61 |

**Table 1** (*continued*)

| Pathogens/ microbiota | Variable (Wash) | Concentration | | | | | | | | | | | |
|---|---|---|---|---|---|---|---|---|---|---|---|---|---|
| | | 3.125 | 1.563 | 0.781 | 0.391 | 0.195 | 0.098 | 0.049 | 0.024 | 0.012 | 0.006 | 0.003 | 0.0015 |
| | | Percentage of Microbial Inhibition (Mean ± Standard Deviation) | | | | | | | | | | | |
| *L. crispatus* | *A. paniculata* intimate wash | −48.15 ± 125.71 | 92.59 ± 31.42 | 187.04 ± 39.28 | 133.34 ± 73.33 | 103.70 ± 0.00 | 83.34 ± 2.62 | 44.45 ± 10.47 | 137.04 ± 52.38 | 7.41 ± 73.33 | 142.60 ± 81.18 | 70.37 ± 5.23 | 114.82 ± 36.66 |
| | SebaMed | 9.52 ± 0.00 | −4.76 ± 0.00 | −23.81 ± 0.00 | −40.48 ± 3.37 | −47.62 ± 3.37 | −47.62 ± 16.84 | −35.72 ± 3.37 | −35.72 ± 10.10 | −35.72 ± 16.84 | −18.17 ± 28.19 | −21.43 ± 10.10 | −19.05 ± 20.20 |
| | Sumber Ayu | −63.16 ± 7.45 | −71.06 ± 7.45 | −78.95 ± 7.45 | −75.00 ± 1.87 | −73.68 ± 0.00 | −65.79 ± 0.00 | −57.90 ± 3.73 | −292.11 ± 331.23 | −38.16 ± 39.07 | −60.53 ± 0.00 | −73.69 ± 3.73 | −63.16 ± 18.61 |
| | Lactacyd | −185.72 ± 10.10 | −371.43 ± 0.00 | −450.00 ± 0.00 | −414.29 ± 0.00 | −271.43 ± 0.00 | −307.14 ± 30.31 | −264.29 ± 20.20 | −264.29 ± 30.30 | −307.14 ± 30.31 | −300.00 ± 70.71 | −257.14 ± 30.31 | −221.43 ± 10.10 |
| | Good Virtues & Co. | −88.89 ± 19.64 | −66.67 ± 3.92 | −141.67 ± 0.00 | −147.23 ± 23.57 | −94.44 ± 0.00 | −100.00 ± 7.86 | −119.45 ± 7.86 | −69.45 ± 3.92 | −50.00 ± 11.78 | −41.67 ± 0.00 | −55.56 ± 0.00 | −52.78 ± 0.00 |

**Notes.**

The results are presented as mean ± standard deviation of microbial inhibition (%), negative values indicate growth promotion.

*C. albicans*, Good Virtues & Co. exhibited the highest inhibition (81.32 ± 3.67%), followed by SebaMed® (64.53 ± 10.78%) at 0.781 mg/ml, Lactacyd® (51.42 ± 5.32%) and Sumber Ayu (41.95 ± 0.20%) at 3.125 mg/ml. For *E. coli*, all the commercial intimate washes showed a similar percentage of inhibition at the highest concentration. SebaMed® demonstrated the highest inhibition (52.40 ± 1.79%), followed by Lactacyd® (44.28 ± 1.10%), Sumber Ayu (43.59 ± 1.05%) and Good Virtues & Co. (41.85 ± 1.16%). SebaMed® (97.31 ± 1.20%) and Lactacyd® (96.12 ± 0.24%) were effective against *S. agalactiae* at the highest concentration, while Sumber Ayu (90.07 ± 1.48%) and Good Virtues & Co. (87.88 ± 11.34%) showed lower inhibition. Certain results were also recorded as negative values. In this case, Lactacyd® highly promoted the growth of *C. albicans* (−9.03 ± 7.84%) at 0.098 mg/ml, followed by SebaMed® at 0.195 mg/ml (−5.40 ± 24.28%). Sumber Ayu promoted the growth of *E. coli* (−4.44 ± 5.52%) at 0.0015 mg/ml and Lactacyd® promoted the growth of *E. coli* (−4.06 ± 5.15%) at 0.012 mg/ml.

The results of pairwise comparison (Table 2) confirmed that *A. paniculata* intimate wash demonstrated significantly higher antimicrobial activity than all commercial intimate washes for *C. albicans* (101.732 to 120.821 ± 3.452%, $p < 0.05$). SebaMed® also showed statistically significant differences compared to Sumber Ayu (6.80 ± 3.45%, $p < 0.05$), Lactacyd® (9.76 ± 3.45%, $p < 0.05$) and Good Virtues & Co. (9.30 ± 3.45%, $p < 0.05$). For *E. coli*, only *A. paniculata* intimate wash showed statistically higher inhibition (67.26 to 74.58 ± 6.41%, $p < 0.05$), while there were no significant differences among the intimate washes themselves. For *S. agalactiae*, significant differences ($p < 0.05$) in inhibition percentages were observed among most intimate washes. However, no significant difference was found between SebaMed® and Lactacyd® (1.123 ± 0.73%, $p > 0.05$), and between Sumber Ayu and Good Virtues & Co. (0.70 ± 0.73%, $p > 0.05$).

In this study, large effect sizes ($\eta^2 p$) were observed across all tested pathogens. For *S. agalactiae*, inhibitions accounted for 97.9% of the variance ($\eta^2 p = 0.979$), and the interaction with intimate washes explained 95.0% ($\eta^2 p = 0.950$). For *C. albicans*, inhibitions and the interaction contributed 71.0% and 72.5% of the variance, respectively ($\eta^2 p = 0.710$ and 0.725). For *E. coli*, these values were 70.1% and 63.5% ($\eta^2 p = 0.701$ and 0.635).

The significance of these effects remained after applying sphericity corrections, indicating robustness. While partial eta squared does not directly provide confidence intervals, the high $F$-values and consistently low $p$-values (all $p < 0.001$) reinforcing the precision and reliability of these strong observed effects. Thus, these findings confirmed both statistical significance and practical relevance of *A. paniculata*'s enhanced antimicrobial effect.

### Effect of intimate washes on *L. crispatus*

*A. paniculata* showed the highest percentage of inhibition against *L. crispatus* (187.04 ± 39.28%) at 0.781 mg/ml. At the highest concentration, it showed to promote growth (−48.15 ± 125.71%). SebaMed® showed the highest growth promotion at 0.195 mg/ml (−47.62 ± 3.37%), while Sumber Ayu and Lactacyd® consistently promoted growth with Lactacyd® showing the highest promotion (−450.00 ± 0.00%). Good Virtues & Co. has also been shown to promote growth (−147.23 ± 23.57%).

**Table 2 Pairwise comparisons of microbial inhibition percentage of intimate washes against vaginal pathogens and microbiota, *L. crispatus*.**

| Pathogens/ microbiota | (I) Wash | (J) Wash | Mean difference (I-J) | Std. error | *p*-value | 95% confidence interval for difference | |
|---|---|---|---|---|---|---|---|
| | | | | | | Lower bound | Upper bound |
| *C. albicans* | | SebaMed | 111.09* | 3.45 | <0.001 | 98.73 | 123.46 |
| | *A. paniculata* intimate wash | Sumber Ayu | 117.89* | 3.45 | <0.001 | 105.53 | 130.25 |
| | | Lactacyd | 120.85* | 3.45 | <0.001 | 108.49 | 133.22 |
| | | Good Virtues & Co. | 101.79* | 3.45 | <0.001 | 89.43 | 114.16 |
| | | *A. paniculata* intimate wash | −111.09* | 3.45 | <0.001 | −123.46 | −98.73 |
| | SebaMed | Sumber Ayu | 6.80 | 3.45 | 0.773 | −5.57 | 19.16 |
| | | Lactacyd | 9.76 | 3.45 | 0.180 | −2.61 | 22.12 |
| | | Good Virtues & Co. | −9.30 | 3.45 | 0.225 | −21.67 | 3.06 |
| | | *A. paniculata* intimate wash | −117.89* | 3.45 | <0.001 | −130.25 | −105.53 |
| | Sumber Ayu | SebaMed | −6.80 | 3.45 | 0.773 | −19.16 | 5.57 |
| | | Lactacyd | 2.96 | 3.45 | 1.000 | −9.40 | 15.33 |
| | | Good Virtues & Co. | −16.10* | 3.45 | 0.009 | −28.46 | −3.73 |
| | | *A. paniculata* intimate wash | −120.85* | 3.45 | <0.001 | −133.22 | −108.47 |
| | Lactacyd | SebaMed | −9.76 | 3.45 | 0.180 | −22.12 | 2.61 |
| | | Sumber Ayu | −2.96 | 3.45 | 1.000 | −15.33 | 9.40 |
| | | Good Virtues & Co. | −19.06* | 3.45 | 0.003 | −31.42 | −6.70 |
| | | *A. paniculata* intimate wash | 101.79* | 3.45 | <0.001 | −114.16 | −89.43 |
| | Good Virtues & Co. | SebaMed | 9.30 | 3.45 | 0.225 | −3.06 | 21.67 |
| | | Sumber Ayu | 16.10* | 3.45 | 0.009 | 3.73 | 28.46 |
| | | Lactacyd | 19.06* | 3.45 | 0.003 | 6.70 | 31.42 |
| *E. coli* | | SebaMed | 67.26* | 6.41 | <0.001 | 44.32 | 90.21 |
| | *A. paniculata* intimate wash | Sumber Ayu | 74.58* | 6.41 | <0.001 | 51.63 | 97.52 |
| | | Lactacyd | 74.35* | 6.41 | <0.001 | 51.40 | 97.29 |
| | | Good Virtues & Co. | 72.91* | 6.41 | <0.001 | 49.97 | 95.86 |
| | | *A. paniculata* intimate wash | −67.26* | 6.41 | <0.001 | −90.21 | −44.32 |
| | SebaMed | Sumber Ayu | 7.31 | 6.41 | 1.000 | −15.63 | 30.26 |
| | | Lactacyd | 7.08 | 6.41 | 1.000 | −15.86 | 30.02 |
| | | Good Virtues & Co. | 5.65 | 6.41 | 1.000 | −17.30 | 28.59 |
| | | *A. paniculata* intimate wash | −74.58* | 6.41 | <0.001 | −97.52 | −51.63 |
| | Sumber Ayu | SebaMed | −7.31 | 6.41 | 1.000 | −30.26 | 15.63 |
| | | Lactacyd | −0.23 | 6.41 | 1.000 | −23.17 | 22.71 |
| | | Good Virtues & Co. | −1.66 | 6.41 | 1.000 | −24.61 | 21.28 |
| | | *A. paniculata* intimate wash | −74.35* | 6.41 | <0.001 | −97.29 | −51.40 |
| | Lactacyd | SebaMed | −7.08 | 6.41 | 1.000 | −30.02 | 15.86 |
| | | Sumber Ayu | 0.23 | 6.41 | 1.000 | −22.71 | 23.17 |
| | | Good Virtues & Co. | −1.43 | 6.41 | 1.000 | −24.38 | 21.51 |

**Table 2** (*continued*)

| Pathogens/ microbiota | (I) Wash | (J) Wash | Mean difference (I-J) | Std. error | *p*-value | 95% confidence interval for difference | |
|---|---|---|---|---|---|---|---|
| | | | | | | Lower bound | Upper bound |
| | Good Virtues & Co. | *A. paniculata* intimate wash | −72.91* | 6.41 | <0.001 | −95.86 | −49.97 |
| | | SebaMed | −5.65 | 6.41 | 1.000 | −28.59 | 17.30 |
| | | Sumber Ayu | 1.66 | 6.41 | 1.000 | −21.28 | 24.61 |
| | | Lactacyd | 1.43 | 6.41 | 1.000 | −21.51 | 24.38 |
| *S. agalactiae* | *A. paniculata* intimate wash | SebaMed | 105.65* | 0.73 | <0.001 | 103.02 | 108.27 |
| | | Sumber Ayu | 110.78* | 0.73 | <0.001 | 108.16 | 113.41 |
| | | Lactacyd | 104.52* | 0.73 | <0.001 | 101.90 | 107.15 |
| | | Good Virtues & Co. | 110.08* | 0.73 | <0.001 | 107.46 | 112.71 |
| | SebaMed | *A. paniculata* intimate wash | −105.65* | 0.73 | <0.001 | −108.27 | −103.02 |
| | | Sumber Ayu | 5.13* | 0.73 | <0.001 | 2.51 | 7.76 |
| | | Lactacyd | −1.13 | 0.73 | 1.000 | −3.75 | 1.50 |
| | | Good Virtues & Co. | 4.43* | 0.73 | 0.001 | 1.81 | 7.06 |
| | Sumber Ayu | *A. paniculata* intimate wash | −110.78* | 0.73 | <0.001 | −113.41 | −108.16 |
| | | SebaMed | −5.13* | 0.73 | <0.001 | −7.76 | −2.51 |
| | | Lactacyd | −6.26* | 0.73 | <0.001 | −8.89 | −3.64 |
| | | Good Virtues & Co. | −0.70 | 0.73 | 1.000 | −3.33 | 1.92 |
| | Lactacyd | *A. paniculata* intimate wash | −104.52* | 0.73 | <0.001 | −107.15 | −101.90 |
| | | SebaMed | 1.13 | 0.73 | 1.000 | −1.50 | 3.75 |
| | | Sumber Ayu | 6.26* | 0.73 | <0.001 | 3.64 | 8.89 |
| | | Good Virtues & Co. | 5.56* | 0.73 | <0.001 | 2.94 | 8.18 |
| | Good Virtues & Co. | *A. paniculata* intimate wash | −110.08* | 0.73 | <0.001 | −112.71 | −107.46 |
| | | SebaMed | −4.43* | 0.73 | 0.001 | −7.06 | −1.81 |
| | | Sumber Ayu | 0.70 | 0.73 | 1.000 | −1.92 | 3.33 |
| | | Lactacyd | −5.56* | 0.73 | <0.001 | −8.18 | −2.94 |
| *L. crispatus* | *A. paniculata* intimate wash | SebaMed | 115.76* | 11.95 | 0.002 | 58.70 | 172.81 |
| | | Sumber Ayu | 173.47* | 11.95 | <0.001 | 116.42 | 230.53 |
| | | Lactacyd | 390.23* | 11.95 | <0.001 | 333.18 | 447.29 |
| | | Good Virtues & Co. | 174.69* | 11.95 | <0.001 | 117.64 | 231.75 |
| | SebaMed | *A. paniculata* intimate wash | −115.76* | 11.95 | 0.002 | −172.81 | −58.70 |
| | | Sumber Ayu | 57.72* | 11.95 | 0.048 | 0.66 | 114.77 |
| | | Lactacyd | 274.48* | 11.95 | <0.001 | 217.42 | 331.53 |
| | | Good Virtues & Co. | 58.94* | 11.95 | 0.044 | 1.88 | 115.99 |
| | Sumber Ayu | *A. paniculata* intimate wash | −173.47* | 11.95 | <0.001 | −230.53 | −116.42 |
| | | SebaMed | −57.72* | 11.95 | 0.048 | −114.77 | −0.66 |
| | | Lactacyd | 216.76* | 11.95 | <0.001 | 159.71 | 273.82 |
| | | Good Virtues & Co. | 1.22 | 11.95 | 1.000 | −55.84 | 58.27 |
| | | *A. paniculata* intimate wash | −390.23* | 11.95 | <0.001 | −447.29 | −333.18 |
| | | SebaMed | −274.48* | 11.95 | <0.001 | −331.53 | −217.42 |
| | | Sumber Ayu | −216.76* | 11.95 | <0.001 | −273.82 | −159.71 |

**Table 2** (*continued*)

| Pathogens/ microbiota | (I) Wash | (J) Wash | Mean difference (I-J) | Std. error | *p*-value | 95% confidence interval for difference | |
|---|---|---|---|---|---|---|---|
| | | | | | | Lower bound | Upper bound |
| | Lactacyd | | | | | | |
| | Good Virtues & Co. | Good Virtues & Co. | −215.54* | 11.95 | <0.001 | −272.60 | −158.49 |
| | | *A. paniculata* intimate wash | −174.69* | 11.95 | <0.001 | −231.75 | −117.64 |
| | | SebaMed | −58.94* | 11.95 | 0.044 | −115.99 | −1.88 |
| | | Sumber Ayu | −1.22 | 11.95 | 1.000 | −58.27 | 55.84 |
| | | Lactacyd | 215.54* | 11.95 | <0.001 | 158.49 | 272.60 |

**Notes.**
*indicates a statistically significant mean difference based on General Linear Model (two-way ANOVA with Bonferroni *post hoc* test), with $p < 0.05$.

For the pairwise comparison (Table 2), only Sumber Ayu and Good Virtues & Co. showed no significant difference ($p > 0.05$). The General Linear Model for repeated measures also indicated that the inhibition level was not statistically significant ($p = 0.296$), although it showed a moderate effect size (partial $\eta^2 = 0.196$). However, the interaction between inhibition levels and intimate washes were statistically significant ($p < 0.001$), with a large effect size (partial $\eta^2 = 0.669$), suggesting that the impact of inhibition varied meaningfully depending on the intimate wash condition. This increases the need to balance pathogen inhibition with microbiota preservation.

## MIC determination of intimate washes

The minimum concentration of *A. paniculata* intimate wash required to inhibit *S. agalactiae* was below 0.0015 mg/ml, as all the tested concentrations showed clear wells even at very low concentrations. In contrast, the MIC for the other pathogens were observed at the highest concentrations (Table 3).

The MIC for the commercial intimate washes on most of the tested pathogens were greater than 3.125 mg/ml. These were shown by the MIC for SebaMed®, Sumber Ayu and Lactacyd® against *C. albicans* and *E. coli* which were >3.125 mg/ml. *S. agalactiae* was recorded an MIC at 1.563 mg/ml when treated with SebaMed® and Lactacyd®, and 3.125 mg/ml when treated with Sumber Ayu and Good Virtues & Co. (Table 3). These findings suggest that *A. paniculata* is a more efficient inhibitor at lower concentrations than other intimate washes.

## MBC determination of *A. paniculata* intimate wash

MBC for *S. agalactiae* was not detected at all concentrations, which indicated that its MBC was less than 0.0015 mg/ml, while *C. albicans* and *E. coli* had MBC at 3.125 mg/ml (Table 3). In this case, *A. paniculata* intimate wash should be diluted to lower than 0.0015 mg/ml. This demonstrates not only inhibition but also a bactericidal effect of *A. paniculata*, especially against *S. agalactiae*, highlighting its strong practical potential for intimate wash formulation.

**Table 3  MIC and MBC values for each intimate wash against vaginal pathogens.**

| Products | Pathogens | MIC (mg/ml) | MBC (mg/ml) |
|---|---|---|---|
| *A. paniculata* intimate wash | *C. albicans* | 3.125 | 3.125 |
| | *E. coli* | 3.125 | 3.125 |
| | *S. agalactiae* | <0.0015 | <0.0015 |
| SebaMed Feminine Wash | *C. albicans* | >3.125 | NT |
| | *E. coli* | >3.125 | NT |
| | *S. agalactiae* | 1.563 | NT |
| Sumber Ayu | *C. albicans* | >3.125 | NT |
| | *E. coli* | >3.125 | NT |
| | *S. agalactiae* | 3.125 | NT |
| Lactacyd | *C. albicans* | >3.125 | NT |
| | *E. coli* | >3.125 | NT |
| | *S. agalactiae* | 1.563 | NT |
| Good Virtues & Co. | *C. albicans* | 1.563 | NT |
| | *E. coli* | >3.125 | NT |
| | *S. agalactiae* | 3.125 | NT |

**Notes.**

MIC values were determined by visual turbidity observation while MBC values were determined as the lowest concentration of the intimate wash that resulted in a 99.9% reduction in viable bacterial count. MBC testing was not conducted for commercial intimate washes as it was assumed that such testing had been performed by the manufacturers prior to product commercialization. NT represents as not tested.

## DISCUSSION

### Broth microdilution method for antimicrobial activity

In this study, the broth microdilution method was applied as it is fast, affordable and can be easily used (*Tuan Kub, Ab Manaf & Abdul Salim, 2021*). In this method, serial dilutions of compounds in broth are mixed with microbial suspensions and incubated overnight. Turbidity is measured, where lower turbidity indicates stronger inhibition. A medium without antimicrobials serves as the control (*Bubonja-Šonje, Knezević & Abram, 2020*). This method is more appropriate compared to disc diffusion and agar well diffusion methods. This is because the active compounds in the formulation may not diffuse effectively from discs into the agar medium, leading to reduced observable antimicrobial effects. Water-insoluble substances can precipitate within the disc, hindering the release of active agents into the agar (*Fernández-Torres et al., 2006*; *Tuan Kub, Ab Manaf & Abdul Salim, 2021*). Although the agar well diffusion method can demonstrate effective zones of inhibition, it does not provide the MIC required to determine the final working concentration of the antimicrobial agent (*Ismail et al., 2021*).

### Analysis of microbial inhibition by *A. paniculata* and commercial intimate washes

Increasing the concentration of *A. paniculata* intimate wash showed a slight rise in *E. coli* inhibition, similar to SebaMed® and Lactacyd®, which also exhibited increased inhibition against all tested pathogens. However, inhibition varied among pathogens regardless of

dilution, as seen with the consistent inhibition of *S. agalactiae* by *A. paniculata* intimate wash at all concentrations. This variability may result from antagonistic interactions between MHB components and active compounds in the intimate washes, altering their effective antimicrobial concentrations which is an issue common with natural extracts (*Caesar & Cech, 2019*). The fractional inhibitory concentration index measures how different agents work together, showing synergy when their combined effects are stronger than expected, and antagonism when their combined effectiveness is lower (*Guzman et al., 2013*; *Singh et al., 2011*; *Roell, Reif & Motsinger-Reif, 2017*). Various antimicrobials with different mechanisms were tested to capture a broad response.

The findings showed that andrographolide had appropriate synergistic effects with preservatives, surfactants and solubilizers against all the tested pathogens. Several benefits might come from the combination of antimicrobial agents, including increased antibacterial activity, lowered dose-dependent side effects, a shorter duration of long-term antimicrobial therapy, and a reduction in the emergence of resistant microorganisms (*Wagner & Ulrich-Merzenich, 2009*). However, it is essential to consider pH, surfactant content, and preservative type as these factors may contribute to variability in inhibition zones and must be accounted for when interpreting comparative antimicrobial data.

A study by *Tuan Kub, Ab Manaf & Abdul Salim (2021)* stated that a formulation is considered effective if it achieves over 90% microbial inhibition. In this study, the percentage of microbial inhibition might be related to the turbidity of MIC. According to all the results, *A. paniculata* intimate wash showed better antimicrobial activities against all the tested pathogens compared to other commercial intimate washes. SebaMed®, Sumber Ayu and Lactacyd® showed potential good inhibition towards *S. agalactiae* with the percentage of microbial inhibition more than 90%. However, negative values suggest the treatment may promote microbial growth by stimulating metabolism or acting as a nutrient source as a value of −100% indicates bacterial activity doubled compared to the control (*Larsen, Zylka & Scott, 2009*). This unexpected growth may be due to disturbances in pH levels, which potentially shifts the pH away from the optimal acidic range (*Lin et al., 2021*). Increased activity might also involve responses from natural vaginal flora rather than pathogens alone, highlighting the importance of considering both flora and media composition in interpreting results (*Bonnet et al., 2019*).

Aloe vera is the active ingredient in SebaMed® and in Lactacyd®. According to *Nabila & Putra (2020)*, aloe vera ethanol extract (6.25%) showed concentration-dependent antifungal activity against *C. albicans*. In Sumber Ayu, *Piper betle* is the only identifiable active ingredient, however, *Gloria et al. (2021)* reported that a formulation combining 25% betel leaf and 50% citrus lime extract showed the lowest inhibition against *E. coli*. Lactacyd® also contains *Rosmarinus officinalis* and *Salvia officinalis*, which are both known antifungal effects against *C. albicans* (*Nabila & Putra, 2020*; *Pawłowska, Janda & Jakubczyk, 2020*; *Ahangari et al., 2019*). Good Virtues & Co. also includes *Calendula officinalis* and *Nigella sativa*, which have shown inhibitory effects on *C. albicans* and *S. agalactiae*. *N. sativa* has strong antifungal activity and can reduce *C. albicans* with or without fluconazole (*Rusda et al., 2020*), while *C. officinalis* is effective for BV treatment (*Pazhohideh et al.,*

*2018*). However, in this study, Sumber Ayu showed low inhibition against *C. albicans* at 3.125 mg/ml, and Good Virtues & Co. showed lower inhibition against *E. coli*.

*Dafur et al. (2024)* reported that andrographolide was effective against *E. coli* but ineffective against *C. albicans*. In contrast, this study showed that the percentage of microbial inhibition of *C. albicans* is greater than *E. coli*. These could be due to some intimate washes that might contain more mild cleansing agents including surfactant and preservatives or the low concentration of active ingredients to inhibit the growth of pathogens. It could also result from counteraction, which describes an unfavourable outcome where the combination of herbs produced toxic or severe side effects (*Che et al., 2013*). Therefore, further research on the synergistic effects of herbs is needed to understand how these examples relate to one another.

## MIC of *A. paniculata* and commercial intimate washes

The highest MIC should be recorded when a single skipped well happens during a microdilution test (*Cockerill et al., 2008*). *A. paniculata* extract has shown antimicrobial activity with MIC ranging from 7.80 to 250 µg/ml against oral pathogens (*Krithigaa et al., 2023*) and 2 to 3 mg/ml against *E. coli*, *P. aeruginosa*, and *S. aureus* (*Srivastava et al., 2024*). It can be concluded that *A. paniculata* extract and other active ingredients in commercial intimate washes exhibit antimicrobial activity, with MIC values varying depending on the specific microorganism tested.

The MIC of *A. paniculata* against *S. agalactiae* was <0.0015 mg/ml, but the exact value could not be determined as no further dilutions have been conducted. The standard broth microdilution method is not optimized for such low concentrations, especially with crude extracts. The high sensitivity may be due to flavonoids and terpenes which can disrupt biofilm formation and modulate virulence factors and efflux pumps (*Pérez-Flores et al., 2025*). The structural vulnerability of *S. agalactiae*, due to its lack of an outer membrane, may further contribute to this response.

## MBC for *A. paniculata* intimate wash

The antimicrobial broth dilution that stops the organism from growing on the agar is called the MBC (*Sykes & Rankin, 2013*). A bactericidal effect was defined as killing 99.9% of the bacteria (*Heuser, Becker & Idelevich, 2022*). *Barata et al. (2023)* reported that *A. paniculata* mouthwash showed MBCs for all tested bacteria except *S. aureus*, with values of 62.5 mg/ml for *Streptococcus mutans* and 15.63 mg/ml for *Streptococcus sobrinus*. The close alignment of MIC and MBC values indicated strong bactericidal activity.

In the present study, while *S. agalactiae* showed a low MIC, MBC testing was limited to the first three wells. For accurate determination, subculturing of all dilution levels is recommended. MBC testing was only conducted for the *A. paniculata* intimate wash to support herbal product development. This was not done for commercial intimate washes, as MIC testing showed no inhibitory activity even at the highest concentration tested.

## Summary of antimicrobial efficacy: percentage inhibition, MIC, and MBC results

These findings confirmed the hypothesis that *A. paniculata* intimate wash demonstrated significant antimicrobial activity compared to commercial intimate washes. All tested concentrations showed strong inhibition against all tested pathogens. Compared to other commercial intimate washes, Good Virtues & Co. showed better inhibition against *C. albicans* at 3.125 mg/ml but did not reach effective antimicrobial thresholds. While other brands effectively inhibited *S. agalactiae*, Good Virtues & Co. showed lower effects at the same concentration. Although clinical data on *A. paniculata* at 3.125 mg/ml in intimate washes is limited, few studies support its safety and efficacy at higher concentrations in mucosal applications.

*Alojid et al. (2021)* reported strong antimicrobial effects of *A. paniculata* extract against oral pathogens at 1.0 g/ml, and a follow-up study confirmed the efficacy and safety of *A. paniculata* mouthwash. Cytotoxicity tests showed non-toxic $LC_{50}$ and $IC_{50}$ values (3.255 mg/ml and 43.55 mg/ml, respectively) with no harmful heavy metal content (*Alojid et al., 2022*). These findings suggest that the 3.125 mg/ml concentration used is within a safe and practical range for mucosal application.

## Effect of *A. paniculata* and commercial intimate washes onto *L. crispatus*

Maintaining vaginal health depends on *Lactobacillus* species, which produce lactic acid to sustain low pH, inhibit pathogens, and rely on glycogen from estrogenized cells (*Pendharkar et al., 2023*; *Spear et al., 2014*). Disruption of this balance can lead to gynecologic issues (*Łaniewski, Ilhan & Herbst-Kralovetz, 2020*). Thus, safe and effective formulations are essential. A 28-day study found lactic acid-containing washes improved moisturization without altering pH (*Bruning et al., 2020*). In contrast, harsh surfactants and parabens may harm microbiota, leading to irritation and infection risk (*Fashemi et al., 2013*; *Łaniewski et al., 2021*).

In this study, *A. paniculata* intimate wash at its MIC effectively inhibited vaginal pathogens while promoting the growth of *L. crispatus*, making it the optimal concentration for further development. Most commercial intimate washes showed growth promotion for *L. crispatus* at all concentrations, except SebaMed®, which inhibited growth only at higher concentrations. No MIC was detected for commercial intimate washes against pathogens, possibly due to the use of diluted products.

MIC was not determined for *L. crispatus* due to limitations with standard media and VC. Future MIC testing for *L. crispatus* should consider using De Man–Rogosa–Sharpe broth or sodium chloride solution, with spectrophotometric or colorimetric assays for improved accuracy (*Mahmoud et al., 2023b*; *Hulankova, 2024*). While using different approaches, this study aligns with *Kyser et al. (2023)*, who developed 3D-bioprinted *L. crispatus* scaffolds for localized probiotic delivery to suppress *G. vaginalis*. Both studies support microbiome-conscious, site-specific strategies to maintain vaginal microbial balance and prevent dysbiosis.

## Study limitations and potential for improvement

The potential variability in the results could be due to the limited number of replicates used in this study. Increasing the number of replications could improve the reliability and statistical power of the findings. Additionally, one of the disadvantages of the broth microdilution global reference method is the challenge posed by the small volume of the test sample, which can affect the accuracy and consistency of the results (*Koeth & Miller, 2023*). The observed growth promotion may be influenced by a decrease in pH at the initial dilution stage. It is recommended that future studies should measure and record the pH after dilution.

In this study, the primary objective was to perform a comparative evaluation of the antimicrobial activity between *A. paniculata* and commercial intimate washes. As the study was designed as an initial screening rather than a confirmatory assay, the focus was placed on relative performance rather than benchmarking against standard antimicrobial agents. Therefore, a positive control such as fluconazole was not included. Products containing water and organic or inorganic compounds require preservation against microbial contamination to ensure consumer safety and extend shelf life. Microbiological safety aims to protect consumers from potentially pathogenic microorganisms while also preventing biological and physicochemical deterioration (*Halla et al., 2018*). Therefore, the stability test of the intimate wash formulation is necessary to assess its long-term effectiveness and safety.

Future research could also be addressed by testing additional pathogens, including drug-resistant strains (*Sunkavalli, McClure & Genco, 2022*). Other than intimate wash formulation, *A. paniculata* can be explored for use in hand wash gel formulations as well. In a study by *Mishra et al. (2022)*, an herbal hand wash gel was developed using an alcoholic extract of *A. paniculata*. The formulation incorporated gelling agent, neutralizer, and surfactant, and the antibacterial activity was evaluated against *S. aureus*, *E. coli*, and *Bacillus subtilis*. The results demonstrated notable antibacterial activity, particularly against *B. subtilis*.

Safety evaluation is crucial in developing vaginal products, especially for regular or clinical use. Although *A. paniculata* is known for its antimicrobial and anti-inflammatory effects, its safety for vaginal application remains underexplored, with no clinical data on *A. paniculata*-based intimate washes. Further testing using relevant *in vitro* models, such as the VK2/E6E7 vaginal epithelial cell line, is essential for assessing cytotoxicity and irritation. For example, *Modi et al. (2013)* found *Rhus parviflora* extract was non toxic to VK2 cells, supporting its safety. Similarly, evaluating *A. paniculata* on VK2 cells is a key step before clinical trials and commercialization.

While cytotoxicity and clinical studies on *A. paniculata* intimate wash are limited, several studies support its potential as a beneficial herbal ingredient due to its antimicrobial, antioxidant, anti-inflammatory, and wound-healing properties. *You et al. (2015)* reported that while *A. paniculata* did not directly stimulate collagen in fibroblasts, it enhanced collagen production when fibroblasts were treated with conditioned medium from epithelium stem cells exposed to the extract, suggesting indirect skin-regenerative and anti-aging effects. Additionally, *Jamaludin et al. (2021)* found that an ethanol-based

*A. paniculata* niosomal gel, which is rich in andrographolide, showed superior antioxidant activity, cytocompatibility, and high wound closure in human fibroblasts. It also promoted re-epithelialization in rats, indicating its promise for safe and effective topical use.

As the widespread use of the *A. paniculata* plant in Malaysia and the demonstrated antimicrobial activity of its intimate wash against tested vaginal pathogens, therefore the plant's potential should be further explored for pharmaceutical applications. The pairwise comparisons showed that the *A. paniculata* intimate wash had higher mean differences in percentage of microbial inhibition compared to other intimate washes. A higher mean difference indicated a stronger antimicrobial activity, suggesting that *A. paniculata* is highly effective against vaginal pathogens due to andrographolide. However, as this research only used diluted commercial intimate washes to make it similar to the diluted *A. paniculata* formulation, it might be the reason why certain commercial intimate washes showed no significant difference in the microbial inhibition compared to the *A. paniculata*. The concentrated form of commercial intimate washes used in daily usage by consumers might show better results (*Barata et al., 2023*).

## CONCLUSIONS

This study explored the antimicrobial potential of *A. paniculata* in an intimate wash formulation as a herbal alternative for vaginal health. The formulated wash showed notable activity against common vaginal pathogens, with 3.125 mg/ml identified as an effective concentration for inhibiting harmful pathogens and supporting beneficial microbiota such as *L. crispatus*. However, as results are based on *in vitro* data, further *in vivo* and clinical studies are needed to confirm safety, efficacy, user tolerance, and potential side effects under real biological conditions.

Among commercial intimate washes, Good Virtues & Co. showed slightly higher inhibition against *C. albicans*, while SebaMed® demonstrated the strongest activity against *S. agalactiae*, indicating its effectiveness against vaginal pathogens. Commercial intimate washes are widely used for maintaining vaginal health and pH balance. For future commercialization of *A. paniculata* formulations, further research on clinical application, toxicity, and bioactive compound characterization is recommended.

### Funding

This work was supported by Universiti Sains Malaysia *via* short term grant (R501-LR-RND002-0000000343-0000) for this study. The funders had no role in study design, data collection and analysis, decision to publish, or preparation of the manuscript.

### Grant Disclosures

The following grant information was disclosed by the authors:
Universiti Sains Malaysia: R501-LR-RND002-0000000343-0000.

## Competing Interests

The authors declare there are no competing interests.

## Author Contributions

- Nur Rina Alissa Razian conceived and designed the experiments, performed the experiments, analyzed the data, prepared figures and/or tables, authored or reviewed drafts of the article, and approved the final draft.
- Tuan Noorkorina Tuan Kub conceived and designed the experiments, analyzed the data, prepared figures and/or tables, and approved the final draft.
- Nurdini Afiqah Suhaimi conceived and designed the experiments, performed the experiments, analyzed the data, prepared figures and/or tables, and approved the final draft.
- Tuan Nadrah Naim Tuan Ismail conceived and designed the experiments, prepared figures and/or tables, and approved the final draft.
- Muhamad Alojid Anis Amiera conceived and designed the experiments, performed the experiments, analyzed the data, prepared figures and/or tables, and approved the final draft.
- Gayus Sale Dafur conceived and designed the experiments, prepared figures and/or tables, and approved the final draft.
- Fatmawati Lambuk conceived and designed the experiments, authored or reviewed drafts of the article, and approved the final draft.
- Rohimah Mohamud conceived and designed the experiments, authored or reviewed drafts of the article, and approved the final draft.
- Ramlah Kadir conceived and designed the experiments, authored or reviewed drafts of the article, and approved the final draft.
- Norzila Ismail conceived and designed the experiments, authored or reviewed drafts of the article, and approved the final draft.
- Norhayati Yusop conceived and designed the experiments, authored or reviewed drafts of the article, and approved the final draft.

## Data Availability

The raw data are available in the Supplemental Files.

## Supplemental Information

Supplemental information for this article can be found online at http://dx.doi.org/10.7717/peerj.20144#supplemental-information.

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
