# Peer review of "Antimicrobial activity of *Androgaphis paniculata* intimate wash and commercial formulations against vaginal pathogens using the broth microdilution method"

_PeerJ, doi:10.7717/peerj.20144_

## Round 0.1 · original submission · Major Revisions

Reviewer 1 ·

Basic reporting

1. Clear and Unambiguous, Professional English Used Throughout
• Some sentences are too long or awkwardly structured, which can affect readability.
• Passive voice overuse in certain sections makes the text unnecessarily wordy.
• Minor grammatical inconsistencies are present (e.g., subject-verb agreement, prepositions, and article usage).
• Some terms need clearer definitions for a broader scientific audience.
Examples of Language Issues & Suggested Revisions:
1. Abstract:
"The purpose of this study is to determine the effectiveness of commercial intimate washes namely SebaMed®, Sumber Ayu, Lactacyd® and Good Virtue & Co. intimate washes together with a formulated Andrographis paniculata intimate wash against vaginal pathogens."
Suggested Revision:
"This study aims to evaluate the antimicrobial efficacy of a newly formulated Andrographis paniculata intimate wash alongside commercial brands (SebaMed®, Sumber Ayu, Lactacyd®, and Good Virtue & Co.) against vaginal pathogens."
2. Introduction:
"A reduction in Lactobacillus spp. can raise the vaginal pH and allow harmful microorganisms to colonize. Vaginal infections may occur when pathogens such as Neisseria gonorrhoeae or Chlamydia trachomatis are exposed during sexual activity or when vaginal dysbiosis enables overgrowth of opportunistic organisms including E. coli and G. vaginalis."
Suggested Revision:
"A decline in Lactobacillus spp. increases vaginal pH, facilitating colonization by pathogenic microorganisms. Vaginal infections may arise from exposure to Neisseria gonorrhoeae or Chlamydia trachomatis during sexual activity or from dysbiosis that allows opportunistic organisms such as E. coli and G. vaginalis to overgrow."
3. Methods:
"The formulated A. paniculata intimate wash was tested alongside commercial products using a broth microdilution method to determine antimicrobial efficacy by calculating microbial inhibition percentages."
Suggested Revision:
"Antimicrobial efficacy of the formulated A. paniculata intimate wash and commercial products was assessed using the broth microdilution method, with microbial inhibition percentages calculated accordingly."

2. Literature References, Sufficient Field Background & Context Provided
Areas for Improvement:
• Some key aspects need more supporting citations:
o Comparative studies on the antimicrobial effectiveness of Andrographis paniculata in intimate washes.
o References discussing why broth microdilution is the preferred method over disk diffusion for evaluating antimicrobial activity.
o More context on how commercial intimate washes influence vaginal microbiota (e.g., pH regulation, surfactant effects).
Suggested Improvements:
• Add references supporting the claim that "vaginal microbiota can change throughout life due to menstruation, douching, antibiotic use, and hormonal fluctuations."
• Include studies that compare synthetic vs. herbal antimicrobial agents to emphasize the novelty of the study.
• Expand on the role of andrographolide, the key bioactive compound in A. paniculata, in antimicrobial mechanisms.
3. Professional Article Structure, Figures, Tables, and Raw Data Shared
Areas for Improvement:
• Figure and Table Labels:
o Some table captions lack detailed explanations of experimental conditions.
From the raw data files, the following tables are referenced in the manuscript but need more detailed captions and explanations of experimental conditions:
Table 1: Microbial Inhibition Percentage of Vaginal Pathogens
 Issue: The table presents inhibition percentages for different intimate washes but lacks a clear explanation of experimental conditions.
 Improvement: Add details on:
The broth microdilution method.
The statistical treatment of inhibition values (mean ± standard deviation).
Why some values are negative (growth promotion).
Table 2: Pairwise Comparison of Antimicrobial Activity
 Issue: The table shows statistical comparisons, but it does not specify:
What statistical test was used (ANOVA, post hoc Tukey test, etc.).
The significance threshold (p-value cutoff).
 Improvement: Clarify the statistical methods used for pairwise comparisons.
Table 3: Minimum Inhibitory Concentration (MIC) Determination
 Issue: The table lists MIC values but does not indicate:
The criteria used to define MIC (visual turbidity, spectrophotometric measurement, or plating).
Why MIC for L. crispatus was not determined.
 Improvement: Clearly state how MIC was determined and whether alternative methods (e.g., MRS broth) should be used for L. crispatus.
Table 4: Minimum Bactericidal Concentration (MBC) of A. paniculata
 Issue: The MBC values are reported, but:
The criteria for MBC determination (e.g., 99.9% bacterial reduction) are missing.
It is unclear why MBC was not determined for commercial washes.
 Improvement:
Add explanations for MBC methodology and why some values are missing.
Ensure axis labels are fully descriptive (e.g., include "mg/ml" for concentration values in MIC/MBC tables).
• Raw Data Consistency:
o Check alignment between raw data and manuscript values.
o Example: The raw data sheet shows MIC for A. paniculata against S. agalactiae as <0.0015 mg/ml, but the manuscript should clarify why the exact MIC value wasn't determined at lower concentrations.
Suggested Improvements:
• Expand figure captions to describe what each figure shows and how it contributes to the research question.
• In tables, standardize decimal places for MIC and inhibition percentages.

4. Self-Contained Manuscript with Relevant Results Linked to Hypotheses
Areas for Improvement:
• Discussion needs stronger integration with hypotheses:
o The manuscript should clearly state:
"These results confirm our hypothesis that A. paniculata intimate wash demonstrates significant antimicrobial activity compared to commercial formulations."
o The current conclusion does not explicitly link results to the hypothesis—revise for clarity.
• Avoid redundancy in results and discussion:
o Some results are repeated in the discussion. Instead, focus on interpreting the findings rather than restating them.
Suggested Improvements:
• Add a concluding statement in the discussion linking results to the hypothesis.
• Remove redundant data points from the discussion that have already been presented in the results section.

Experimental design

1. Original Primary Research within the Aims and Scope of the Journal
Minor Concerns:
• While the study explores an important topic, it does not explicitly state why A. paniculata was chosen over other herbal extracts.
o Suggestion: Briefly mention traditional medicinal uses of A. paniculata in vaginal health.
• The study should clarify how the findings contribute to broader antimicrobial research (e.g., potential for alternative treatments to antibiotics).
Suggested Improvement:
• Add a short section in the introduction or discussion comparing A. paniculata with other known antimicrobial herbal extracts used in similar applications.

2. Research Question Well Defined, Relevant & Meaningful
Areas for Improvement:
• The hypothesis is implied but not explicitly stated.
o The authors should clearly state whether they expect A. paniculata to outperform commercial products.
• The study mentions that commercial washes were diluted before testing, but it is unclear how this impacts real-world efficacy.
Suggested Improvement:
• State the hypothesis explicitly in the introduction:
"We hypothesize that A. paniculata intimate wash will exhibit greater antimicrobial activity against vaginal pathogens while maintaining L. crispatus growth, compared to commercial intimate washes."
• Clarify dilution effects: Does dilution reflect real-use conditions, or was it necessary for standardization?

3. Rigorous Investigation Performed to a High Technical & Ethical Standard
Areas for Improvement:
1. Control groups are not fully explained.
o The study mentions negative controls (NC), viability controls (VC), and sterility controls (SC) but does not explicitly define their function.
o Suggest adding a dedicated subsection on controls in the Methods.
2. Potential bias in data collection.
o The study does not mention blinding during measurements.
o Was there inter-investigator variability, or were tests performed by a single researcher?
Suggested Improvement:
• Clearly define NC, VC, and SC controls and their role in validation.
• Mention who conducted the measurements and whether results were cross-validated by multiple investigators.
4. Methods Described with Sufficient Detail & Information to Replicate
Areas for Improvement:
1. Missing details in the broth microdilution method:
o The manuscript states that percentage inhibition was calculated but does not specify the exact formula used (this was found in the raw data but should be included in the Methods).
o It is unclear whether turbidity readings were taken at multiple time points or only after 24 hours.
2. MIC Determination for L. crispatus:
o The manuscript states that MIC was not determined for L. crispatus due to turbidity issues.
o However, alternative methods exist (e.g., MRS broth for optimal Lactobacillus growth).
o The manuscript should suggest a better approach for future studies.
3. Dilution of Commercial Intimate Washes:
o The study assumes commercial washes are 100% concentrated, then dilutes them.
o This may not reflect real-world conditions (as they are pre-formulated with active ingredients).
o The dilution method should be justified with references.
Suggested Improvements:
• Provide the exact equation used for inhibition calculation.
• Clarify measurement timing in the broth microdilution assay.
• Propose alternative MIC methods for L. crispatus (e.g., different broth medium).
• Justify commercial product dilution and state whether it impacts comparison validity.

Validity of the findings

Validity of the Findings – Detailed Review
1. Impact and Novelty Not Assessed, but Replication is Encouraged
Areas for Improvement:
• The study does not explicitly discuss how future research could validate or expand on the findings.
• The authors should clarify whether they expect similar results in vivo, or if formulation adjustments would be required.
• Why Andrographis paniculata?
o Many antimicrobial herbal extracts exist.
o A brief comparison with other herbal candidates would strengthen the rationale.
Suggested Improvements:
• Add a short section in the Discussion on how the findings could be replicated in other studies (e.g., testing against drug-resistant strains or in different pH conditions).
• Discuss future validation steps, such as clinical trials or in vivo animal models.
• Provide a brief comparison of A. paniculata with other antimicrobial herbal extracts (e.g., Nigella sativa, Curcuma longa).

2. All Underlying Data are Robust, Statistically Sound, and Controlled
Concerns & Areas for Improvement:
1. Negative inhibition values are not fully explained.
o The manuscript reports negative inhibition values (e.g., for C. albicans and E. coli under certain commercial washes).
o It is unclear why some washes promoted microbial growth.
o This needs a stronger biological explanation.
2. Statistical Comparisons Need More Context.
o The manuscript states that A. paniculata was significantly more effective.
o However, it does not discuss:
 Effect size.
 Confidence intervals.
 The practical significance of the statistical differences.
3. Control Groups Need More Justification.
o The study used NC (negative control), VC (viability control), and SC (sterility control).
o It is not fully explained why these controls were chosen and how they validate the findings.
Suggested Improvements:
• Provide a biological explanation for why negative inhibition values occurred.
• Clarify what statistical significance means in practical terms (e.g., does the observed inhibition translate into effective pathogen reduction in real-world use?).
• Clearly state how the control groups validate the findings.

3. Conclusions are Well Stated, Linked to the Original Research Question & Limited to Supporting Results
Concerns & Areas for Improvement:
1. Causation vs. Correlation:
o The study implies causation in some statements, but the results are based on correlation.
o Example:
 "At 3.125 mg/ml, A. paniculata intimate wash exhibited potential effective treatment for selected vaginal pathogens."
 This suggests clinical efficacy, but the study only tested in vitro activity.
2. Limitations Not Fully Addressed.
o The manuscript does not mention potential formulation challenges.
o For example:
 Will the pH of A. paniculata wash affect real-world performance?
 Are there stability concerns in long-term storage?
3. Clinical Application Not Fully Discussed.
o The transition from in vitro to in vivo needs more explanation.
o How does this compare to standard treatments (e.g., clotrimazole, metronidazole)?
Suggested Improvements:
• Revise conclusions to avoid causal claims.
o Instead of:
"This study confirms that A. paniculata is an effective treatment for vaginal pathogens."
o Say:
"This study demonstrates that A. paniculata exhibits antimicrobial properties in vitro, suggesting potential as an intimate wash formulation, pending further in vivo validation."
• Add a short paragraph discussing limitations (e.g., pH stability, shelf life, formulation challenges).
• Compare in vitro findings to existing clinical treatments.

Additional comments

The discussion section could be better structured by using subheadings to organize key points and reducing redundancy, particularly where MIC and MBC values are repeated from the results. The manuscript should also expand on the practical application of Andrographis paniculata as an intimate wash, addressing formulation challenges such as pH stability, long-term storage, and potential skin irritation. Safety considerations should be discussed, including any known adverse effects of A. paniculata on vaginal mucosa and the need for cytotoxicity or clinical testing before commercialization.

Annotated reviews are not available for download in order to protect the identity of reviewers who chose to remain anonymous.

Reviewer 2 ·

Basic reporting

The study present a comparative in vitro study evaluating the antimicrobial efficacy of various commercial intimate washes and a novel Andrographis paniculata-based formulation against common vaginal pathogens, while assessing selectivity toward preserving healthy vaginal flora (Lactobacillus crispatus). The language, literature and structure are appropriate.
The inclusion of L. crispatus as a probiotic marker is a significant strength of this study and should be more prominently emphasized in the introduction and discussion.
The literature, recommended to have form 2022-25. At present no work from 2025. Please refer: https://doi.org/10.1016/j.lwt.2024.116219, https://doi.org/10.1080/19396368.2022.2074325

Experimental design

What were the extraction method, solvents used, concentration standardization, and pH of the final product?

Was the herbal content quantified (e.g., andrographolide content)?

Specify the strain information and sources of each pathogen tested (e.g., ATCC codes or clinical isolates).

Regarding L. crispatus: while turbidity observation was difficult, were there any alternative methods (e.g., plate counts or pH changes) explored for assessing its growth?

Validity of the findings

A table summarizing MIC and MBC values for each product-pathogen pair.

Standard control antibiotics (positive controls) and vehicle controls for benchmarking.

Clarify whether pH, surfactant content, or preservatives in the commercial products might have influenced antimicrobial effects independent of active ingredients.

The findings that A. paniculata showed significant inhibition of E. coli and C. albicans are promising. However, a brief discussion of possible mechanisms of action based on known bioactives in A. paniculata would add scientific value.

Please further discuss the clinical relevance of the concentrations tested (3.125 mg/ml)—is this concentration practical and safe for mucosal application?

---

## Round 0.2 · accepted · Accept

Thanks for addressing all comments!

Reviewer 2 ·

Basic reporting

The author has successfully addressed all the comments and suggestions provided in the previous review. The revisions and editions made are satisfactory and have enhanced the overall quality of the manuscript.

Experimental design

-

Validity of the findings

-